**Data Availability Statement:** The datasets used and/or analyzed during the current study are available from the MIMIC-III and MIMIC-IV

# Longitudinal trajectories of blood glucose and 30-day mortality in patients with diabetes mellitus combined with acute myocardial infarction: A retrospective cohort analysis of the MIMIC database

**Bowen Deng**[1]*, **Qingcheng Liu**[2], **Liang Qiao**[1], **Shun Lv**[1]

1 Department of Traditional Chinese Medicine, Xuhui District Central Hospital, Shanghai, P.R. China,
2 Department of Traditional Chinese Medicine, Jiangpu Community Health Service Center, Shanghai, P.R. China

* dengbowen8693@outlook.com

## Abstract

### Background

Although blood glucose changes have been suggested to be a potential better target for clinical control than baseline blood glucose levels, the association of blood glucose changes with the prognosis in acute myocardial infarction (AMI) patients with diabetes mellitus (DM) is unclear. Herein, this study aimed to investigate association of short-term longitudinal trajectory of blood glucose with 30-day mortality in this population.

### Methods

Data of AMI patients with DM were extracted from the Medical Information Mart for Intensive Care (MIMIC) database in 2003–2019 in this retrospective cohort study. The latent growth mixture modeling (LGMM) model was utilized to classify the 24-hour longitudinal trajectory of blood glucose of the patients. Kaplan-Meier (KM) curve was drawn to show 30-day mortality risk in patients with different trajectory classes. Univariate and multivariate Cox regression analyses were employed to explore the association of longitudinal trajectory of blood glucose within 24 hours after the ICU admission with 30-day mortality. Also, subgroups analysis of age, gender, and AMI types was performed. The evaluation indexes were hazard ratios (HRs) and 95% confidence intervals (CIs).

### Results

Among 1,523 eligible patients, 227 (14.9%) died within 30 days. We identified 4 longitudinal trajectories of blood glucose, including class 1 (a low initial average blood glucose level with steady trend within 24 hours), class 2 (a high initial average blood glucose with gently decreased trend), class 3 (the highest initial average blood glucose with rapidly decreased trend) and class 4 (a high initial average blood glucose level with the trend that increased at first and then decreased). After adjusting for covariates, an average blood glucose level of

database, https://mimic.physionet.org/iii/ and https://mimic.physionet.org/iv/.

**Funding:** The author(s) received no specific funding for this work.

**Competing interests:** The authors have declared that no competing interests exist.

≥200 mg/dL was linked to higher risk of 30-day mortality, comparing to that of <140 mg/dL (HR = 1.80, 95%CI: 1.23–2.63). Comparing to patients whose longitudinal trajectory of blood glucose conformed to class 1, those with class 2 (HR = 2.52, 95%CI: 1.79–3.53) or class 4 (HR = 3.53, 95%CI: 2.07–6.03) seemed to have higher risk of 30-day mortality. Additionally, these associations were also significant in aged ≥60 years old, female, male, NSTEMI, and STEMI subgroups (all *P*<0.05).

## Conclusion

A low level of average blood glucose at the ICU admission or reducing blood glucose to a normal level quickly with adequate measures in 24 hours after ICU admission may be beneficial for AMI patients with DM to reduce the risk of 30-day mortality. These findings may provide some information for further exploration on appropriate range of blood glucose changes in clinical practice.

## Introduction

Acute myocardial infarction (AMI) is the myocardial necrosis attributed to acute and persistent ischemia and hypoxia of coronary arteries, which is a common disease in the intensive care units (ICUs), causing high in-hospital mortality [1]. Diabetes mellitus (DM) increases the risk of AMI [2]. AMI patients with DM had a significantly increased risk of short-, medium-, and long-term mortality compared with those without DM [3]. Therefore, controlling blood glucose in an appropriate range is critical for prognoses in patients with AMI combined with DM.

Epidemiological investigations have shown that elevated blood glucose levels on admission are common in patients with AMI, which is significantly associated with poor prognoses regardless of DM; the baseline blood glucose levels in patients with DM are higher than those in patients without DM [4–6]. Recently, researchers suggested that blood glucose changes may be a better target for clinical control than baseline blood glucose levels. Persistent hyperglycemia during hospitalization was a better predictor for the risk of mortality in patients with AMI than that on admission [7]. A previous study has also found that long-term poor longitudinal trajectories of blood glucose are significantly linked to the risk of mortality, even in persons with normal baseline blood glucose, longitudinal elevation of blood glucose is associated with a higher risk of mortality [8]. In addition, changes in the longitudinal trajectory of blood indicators/vital signs were found to be significantly associated with short-term prognosis in critically ill patients [9, 10]. However, short-term changes in blood glucose trajectories and their impact on prognosis in AMI patients with DM is still unclear.

Herein, this study intended to investigate the association of short-term longitudinal trajectory of blood glucose with 30-day mortality in AMI patients with DM, with the aim of providing some information on finding a reasonable range of short-term blood glucose control in patients with DM combined with AMI, and further on treatment decision and prognosis improvement.

## Methods

### Study design and population

In this retrospective cohort study, data of participants were extracted from the Medical Information Mart for Intensive Care (MIMIC)-III and MIMIC-IV database in 2003–2019. The

MIMIC was jointly published by the Computational Physiology Laboratory of Massachu-setts Institute of Technology (MIT), Beth Israel Deaconess Medical Center (BIDMC) and Philips Medical. This database contains information on true hospital stays for patients admitted to a tertiary academic medical center in Boston, MA, USA, and is intended to support a wide variety of research in healthcare. More details are shown on the MIMIC website: https://mimic. mit.edu/docs/.

There were a total of 3,936 AMI patients with DM in the database. The inclusion criteria were (1) aged ≥18 years old, (2) hospitalized in the ICU for at least 24 hours, and (3) with measurement of blood glucose per 6-hour interval [11]. The exclusion criteria were (1) not received blood glucose measurement and (2) missing survival information. Finally, 1,523 patients were eligible. The requirement of ethical approval was waived by the Institutional Review Board of Xuhui District Central Hospital, because the study data were obtained from MIMIC (a publicly available database). The need for written informed consent was also waived by the Institutional Review Board of Xuhui District Central Hospital due to the retrospective nature of this study.

## Diagnosis of AMI and DM

The diagnosis of AMI at the ICU admission in the MIMIC database was according to the International Classification of Diseases, 9th revision (ICD-9) code (41000 to 41092) or 10th revision (ICD-10) code that begin with I21 (excepting I21A, I21A1, and I21A9) [12]. To be specific, the codes for ST-segment elevation myocardial infarction (STEMI) begin with 4100–4106 and 4108 (ICD-9 codes) or begin with I210-I213 (ICD-10 codes), and the codes for non-STEMI (NSTEMI) begin with 4107 (ICD-9 codes) or begin with I214 (ICD-10 codes). Similarly, the diagnosis of DM was according to the ICD-9 code (begin with 250) or ICD-10 code (begin with E10-E14). Besides, we only extracted the information of multiple-admission patients when they stayed in the hospital for the first time.

## Construction of longitudinal trajectory of blood glucose

In this study, we utilized the latent growth mixture modeling (LGMM) to classify the 24-hour longitudinal trajectory of blood glucose of AMI patients with DM. The LGMM approach is a statistical method that consider individuals being part of a heterogeneous population, composed of unobserved groups of individual trajectories sharing similar characteristics [13]. The goal of latent growth modeling approaches is to estimate a given number of a priori unobserved latent groups within a population, based on the probability of membership of individuals to a specific trajectory group [14].

Regarding to the process of LGMM construction, in brief, it were fitted on the basis of 4 conditions. Firstly, among different numbers of classes, the final selected number of classes should include classes have the smallest values of both the Akaike information criterion (AIC) and the Bayesian information criterion (BIC), whereas have the largest Log likelihood ratio. Secondly, the entropy need to be more than 0.7. Thirdly, the minimum proportion of each class should be not less than 1%. Finally, the average posterior probability of each class should be more than 70%.

In this study, there were 4 different longitudinal trajectories of blood glucose among patients with AMI combined with DM. To be specific, the class 1 represented a low initial average blood glucose level with steady trend within 24 hours (which was set to be the reference); class 2 represented a high initial average blood glucose with gently decreased trend; class 3 had the highest initial average blood glucose with rapidly decreased trend; class 4 showed a high initial average blood glucose level with the trend that increased at first and then decreased.

## Variable selection

We also selected variables as potential covariates from the database, including age, gender, race, AMI types, 48-hour urine output, sepsis, cardiogenic shock (CS), weight, heart rate (HR), systolic blood pressure (SBP), diastolic blood pressure (DBP), respiratory rate (RR), temperature, $SPO_2$, the Sequential Organ Failure Assessment (SOFA) score, the Simplified Acute Physiology Score (SAPS) II, the Glasgow Coma Scale (GCS), the Charlson Comorbidity Index (CCI), white blood cell (WBC), platelet, hemoglobin (HB), red cell distribution width (RDW), creatinine (Cr), pH, international normalized ratio (INR), prothrombin time (PT), blood urea nitrogen (BUN), bicarbonate, sodium (Na), potassium (K), chloride, magnesium (Mg), glucose, ventilation use, vasopressor use, received percutaneous coronary intervention (PCI) or coronary artery bypass graft (CABG), thrombolysis, insulin use, antiplatelet agents use, anticoagulation agents use, and statins use.

The serum blood glucose concentrations were divided into three levels: normal level ($<140$ mg/dL), intermediate level (140–200 mg/dL) and high level ($\geq 200$ mg/dL) [15]. Sepsis was diagnosed according to the sepsis-3 criteria [16], which in brief, patients with documented or suspected infection and an acute change in total SOFA score of $\geq 2$ points were considered to have sepsis. Infection was identified using the ICD codes, including 99591 and 99592 (ICD-9) or A021, A227, A267, A327, codes begin with A40, A41, and R652 (ICD-10). CS was diagnosed basing on the ICD codes (ICD-9 78551 and ICD-10 R570). PCI was defined by the ICD-9 procedure code (0066, 3606, 3607, 3609, and 3610) and ICD-10 procedure code (begin with 02703, 02713, 02723, and 02733); CABG was also defined by the ICD-9 procedure code (3611–3619 and 362) and ICD-10 procedure code (begin with 0210–0213).

Thrombolysis was judged through thrombolytic drug item ID 221319 (Alteplase), ICD-9 procedure code (9910 and 3604), or ICD-10 procedure code (3E07017 and 3E07317). Anticoagulant drug item ID including warfarin (225913), heparin (225152, 229597 and 225975), bivalirudin (225148 and 229781), argatroban (225147), lepirudin (221892), fondaparinux (225908); and antiplatelet drugs item ID including tirofiban (225157). In addition, the statins that recorded in the MIMIC were atorvastatin, fluvastatin, pitavastatin, lovastatin, pravastatin, rosuvastatin and simvastatin.

## Outcome and follow-up duration

The study outcome was 30-day mortality. The MIMIC followed up by information in the electronic medical charts and hospital department records, or making contact with the patients, their family members, their attending health care workers, or family physicians on the phone. The follow-up ended when patients died or 30 days after the ICU admission.

## Statistical analysis

Normally distributed data were described using mean ± standard deviation (Mean ± SD), and analysis of variance (ANOVA) was utilized for the comparison among different trajectory class groups. Frequency and composition ratio [N (%)] was used to describe the distribution of categorical data, and chi-square test ($\chi^2$) or the Fisher exact test were used for comparation.

The LGMM model including 4 classes was established using R package "LCMM" (https://www.jstatsoft.org/article/view/v078i02). The Kaplan-Meier (KM) cumulative incidence curve was draw to show the 30-day mortality risk in patients with different trajectory classes. Univariate Cox regression analysis was utilized to screen covariates, namely variables significantly associated with 30-day mortality (with $P<0.05$), and they were further included in the adjustment of multivariate models. Univariate and multivariate Cox regression analyses were employed to investigate the association of short-term longitudinal trajectory of blood glucose

with 30-day mortality in AMI patients with DM. Model 1 was crude model. Model 2 adjusted for vasopressor use, PCI/CABG, antiplatelet use, anticoagulation agents use, and statins use. Model 3 adjusted for age, AMI types, urine output, sepsis, CS, HR, DBP, RR, $SpO_2$, SOFA, SAPS-II, CCI, WBC, platelet, HB, RDW, Cr, pH, INR, PT, BUN, bicarbonate, Na, chloride, Mg, vasopressor use, PCI/CABG, antiplatelet agents use, anticoagulation agents use, and statins use. We also performed subgroup analysis of age, gender and AMI types to explore the relationship between short-term longitudinal trajectory of blood glucose and 30-day mortality. The evaluation indexes were hazard ratios (HRs) and 95% confidence intervals (CIs). $P<0.05$ indicated significant difference. Statistical analyses used R version 4.2.1. (2022-06-23 ucrt).

## Results

### Selection of trajectory classes of blood glucose in LGMM model

S1 and S2 Tables respectively showed the determination of the number of classes and the average posterior probability of selected classes. We first set the number of classes into 1–5, according to the Log likelihood, AIC, BIC and the proportion of different classes for each condition, it seemed that the LGMM model had the best fitness when divided into 4 classes. In addition, Fig 1 clearly showed the trajectory plot of patients with 4 different trajectory patterns of average blood glucose.

### Characteristics of AMI patients with DM

The characteristics of eligible patients were shown in the Table 1. Among 1,523 AMI patients with DM, 227 (14.9%) died within 30 days. There were respectively 1,098, 235, 140, and 50 patients in class 1 to 4 of trajectory patterns of average blood glucose. The average blood glucose concentration among these four groups were significantly different (class 1: 154.00 mg/dL; class 2: 262.00 mg/dL; class 3: 435.50 mg/dL; class 4: 314.00 mg/dL). In addition, among the 4 blood glucose trajectory groups, age, gender, race, AMI types, 48-hour urine output,

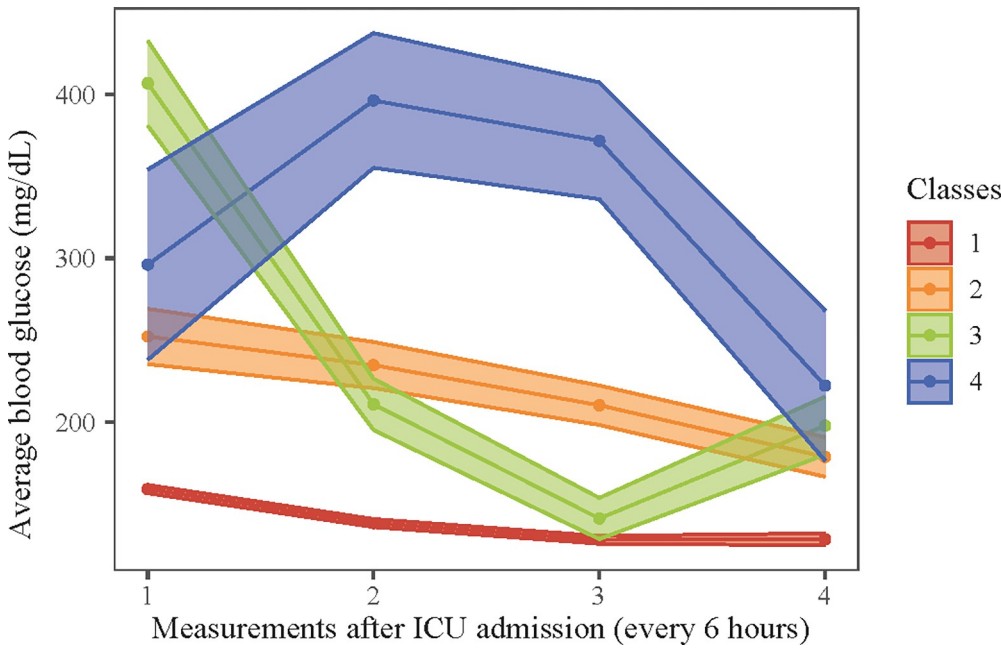

**Fig 1. The trajectory plot of patients with 4 different trajectory patterns of average blood glucose.**

**Table 1. Characteristics of AMI patients with DM.**

| Variables | Total (n = 1523) | Blood glucose trajectory classes | | | | Statistic | P |
|---|---|---|---|---|---|---|---|
| | | 1 (n = 1098) | 2 (n = 235) | 3 (n = 140) | 4 (n = 50) | | |
| Age, years, Mean ± SD | 67.77 ± 11.56 | 68.29 ± 11.06 | 67.23 ± 12.23 | 65.39 ± 13.23 | 65.41 ± 13.18 | F = 3.607 | **0.013** |
| Gender, n (%) | | | | | | $\chi^2 = 11.815$ | **0.008** |
| Female | 536 (35.19) | 361 (32.88) | 89 (37.87) | 64 (45.71) | 22 (44.00) | | |
| Male | 987 (64.81) | 737 (67.12) | 146 (62.13) | 76 (54.29) | 28 (56.00) | | |
| Race, n (%) | | | | | | $\chi^2 = 9.211$ | **0.027** |
| White | 552 (36.24) | 397 (36.16) | 89 (37.87) | 40 (28.57) | 26 (52.00) | | |
| Others | 971 (63.76) | 701 (63.84) | 146 (62.13) | 100 (71.43) | 24 (48.00) | | |
| AMI types, n (%) | | | | | | - | **<0.001** |
| NSTEMI | 1216 (79.84) | 916 (83.42) | 168 (71.49) | 95 (67.86) | 37 (74.00) | | |
| STEMI | 216 (14.18) | 122 (11.11) | 50 (21.28) | 35 (25.00) | 9 (18.00) | | |
| Unknown | 91 (5.98) | 60 (5.46) | 17 (7.23) | 10 (7.14) | 4 (8.00) | | |
| 48-hour urine output, Mean ± SD | 2945.00 (1818.50, 4219.00) | 2950.00 (1907.50, 4171.00) | 2818.00 (1527.50, 4379.50) | 3247.50 (1617.75, 4738.50) | 2104.50 (863.00, 3599.50) | $\chi^2 = 8.053$ | **0.045** |
| Sepsis, n (%) | | | | | | $\chi^2 = 74.439$ | **<0.001** |
| No | 1316 (86.41) | 999 (90.98) | 168 (71.49) | 110 (78.57) | 39 (78.00) | | |
| Yes | 207 (13.59) | 99 (9.02) | 67 (28.51) | 30 (21.43) | 11 (22.00) | | |
| CS, n (%) | | | | | | $\chi^2 = 83.710$ | **<0.001** |
| No | 1331 (87.39) | 1011 (92.08) | 178 (75.74) | 100 (71.43) | 42 (84.00) | | |
| Yes | 192 (12.61) | 87 (7.92) | 57 (24.26) | 40 (28.57) | 8 (16.00) | | |
| Weight, kg, Mean ± SD | 84.50 (72.00, 98.84) | 84.82 (72.35, 98.97) | 85.82 (73.45, 100.00) | 80.00 (67.96, 93.80) | 85.91 (76.57, 96.20) | $\chi^2 = 8.624$ | **0.035** |
| HR, Mean ± SD | 86.71 ± 16.67 | 84.01 ± 14.21 | 91.77 ± 19.21 | 94.81 ± 21.39 | 99.40 ± 20.34 | F = 40.421 | **<0.001** |
| SBP, mmHg, Mean ± SD | 117.47 ± 23.23 | 115.72 ± 21.17 | 119.98 ± 26.32 | 122.16 ± 26.06 | 130.98 ± 33.83 | F = 10.724 | **<0.001** |
| DBP, mmHg, Mean ± SD | 60.20 ± 15.63 | 58.33 ± 14.23 | 64.97 ± 17.15 | 62.99 ± 18.72 | 71.04 ± 18.07 | F = 23.039 | **<0.001** |
| RR, bpm, Mean ± SD | 17.34 ± 5.96 | 16.18 ± 5.38 | 20.08 ± 6.57 | 20.52 ± 5.68 | 21.06 ± 7.07 | F = 55.410 | **<0.001** |
| Temperature, ˚C, Mean ± SD | 36.42 ± 0.84 | 36.37 ± 0.75 | 36.67 ± 0.96 | 36.34 ± 1.06 | 36.57 ± 1.18 | F = 9.551 | **<0.001** |
| $SpO_2$, %, Mean ± SD | 97.77 ± 3.93 | 98.33 ± 3.56 | 96.44 ± 3.90 | 96.06 ± 5.35 | 96.56 ± 4.11 | F = 28.373 | **<0.001** |
| SOFA, Mean ± SD | 3.00 (1.00, 5.00) | 3.00 (1.00, 5.00) | 3.00 (1.00, 4.00) | 2.00 (1.00, 4.00) | 3.00 (0.00, 4.00) | $\chi^2 = 3.492$ | 0.322 |
| SAPS-II, Mean ± SD | 41.00 (32.00, 51.00) | 40.00 (32.00, 49.00) | 46.00 (36.00, 56.00) | 46.00 (35.00, 53.00) | 51.00 (44.00, 56.75) | $\chi^2 = 50.377$ | **<0.001** |
| GCS, Mean ± SD | 15.00 (12.00, 15.00) | 15.00 (10.00, 15.00) | 15.00 (13.50, 15.00) | 15.00 (14.00, 15.00) | 15.00 (15.00, 15.00) | $\chi^2 = 7.980$ | **0.046** |
| CCI, Mean ± SD | 5.77 ± 2.45 | 5.68 ± 2.54 | 6.03 ± 2.23 | 5.89 ± 2.01 | 6.20 ± 2.36 | F = 1.947 | 0.120 |
| WBC, K/uL, Mean ± SD | 12.60 (9.60, 17.00) | 12.10 (9.20, 16.28) | 13.80 (10.55, 19.30) | 14.40 (11.00, 19.30) | 15.60 (10.95, 21.32) | $\chi^2 = 42.533$ | **<0.001** |
| Platelet, K/uL, Mean ± SD | 180.00 (132.00, 244.00) | 165.00 (128.00, 220.00) | 223.00 (152.50, 293.00) | 242.50 (169.00, 323.50) | 236.00 (144.00, 284.50) | $\chi^2 = 111.414$ | **<0.001** |
| HB, g/dL, Mean ± SD | 10.00 (8.50, 11.40) | 9.60 (8.20, 11.10) | 10.80 (9.20, 12.20) | 10.90 (9.30, 12.43) | 10.75 (9.10, 11.70) | $\chi^2 = 72.399$ | **<0.001** |
| RDW, %, Mean ± SD | 14.62 ± 1.86 | 14.59 ± 1.87 | 14.71 ± 1.95 | 14.64 ± 1.74 | 14.88 ± 1.45 | F = 0.588 | 0.623 |
| Cr, mg/dL, Mean ± SD | 1.76 ± 1.80 | 1.58 ± 1.63 | 2.13 ± 2.17 | 2.47 ± 2.20 | 1.97 ± 1.32 | F = 14.807 | **<0.001** |
| pH, Mean ± SD | 7.37 ± 0.10 | 7.39 ± 0.08 | 7.33 ± 0.11 | 7.29 ± 0.13 | 7.29 ± 0.14 | F = 78.821 | **<0.001** |
| INR, Mean ± SD | 1.52 ± 0.67 | 1.52 ± 0.62 | 1.47 ± 0.60 | 1.53 ± 0.95 | 1.73 ± 0.94 | F = 2.097 | 0.099 |
| PT, sec, Mean ± SD | 16.23 ± 5.99 | 16.26 ± 5.76 | 15.69 ± 5.20 | 16.14 ± 7.56 | 18.27 ± 8.61 | F = 2.578 | 0.052 |
| BUN, mg/dL, Mean ± SD | 24.00 (16.00, 39.00) | 20.00 (15.00, 31.00) | 32.00 (20.00, 54.50) | 41.00 (27.00, 58.00) | 35.00 (25.00, 46.00) | $\chi^2 = 181.463$ | **<0.001** |
| Bicarbonate, mEq/L, Mean ± SD | 21.71 ± 4.37 | 22.71 ± 3.37 | 20.12 ± 4.86 | 17.89 ± 6.27 | 18.08 ± 5.00 | F = 90.251 | **<0.001** |
| Na, mEq/L, Mean ± SD | 135.89 ± 4.57 | 135.66 ± 3.90 | 137.04 ± 5.87 | 135.44 ± 6.17 | 136.94 ± 5.04 | F = 7.330 | **<0.001** |
| K, mEq/L, Mean ± SD | 4.57 ± 0.85 | 4.55 ± 0.80 | 4.56 ± 0.98 | 4.68 ± 0.92 | 4.72 ± 1.08 | F = 1.477 | 0.219 |
| Chloride, mEq/L, Mean ± SD | 104.26 ± 6.09 | 104.97 ± 5.40 | 102.95 ± 7.11 | 100.90 ± 7.46 | 104.18 ± 6.98 | F = 23.813 | **<0.001** |

*(Continued)*

**Table 1.** (Continued)

| Variables | Total (n = 1523) | Blood glucose trajectory classes | | | | Statistic | P |
|---|---|---|---|---|---|---|---|
| | | 1 (n = 1098) | 2 (n = 235) | 3 (n = 140) | 4 (n = 50) | | |
| Mg, mg/dL, Mean ± SD | 2.21 ± 0.55 | 2.28 ± 0.56 | 2.02 ± 0.50 | 2.02 ± 0.47 | 2.00 ± 0.38 | F = 24.471 | **<0.001** |
| Glucose, mg/dL, Mean ± SD | 173.00 (137.00, 236.00) | 154.00 (128.00, 185.00) | 262.00 (207.50, 324.00) | 435.50 (334.75, 508.00) | 314.00 (159.25, 451.25) | $\chi^2$ = 630.191 | **<0.001** |
| Ventilation use, n (%) | | | | | | $\chi^2$ = 10.959 | **0.012** |
| No | 472 (30.99) | 324 (29.51) | 91 (38.72) | 47 (33.57) | 10 (20.00) | | |
| Yes | 1051 (69.01) | 774 (70.49) | 144 (61.28) | 93 (66.43) | 40 (80.00) | | |
| Vasopressor use, n (%) | | | | | | $\chi^2$ = 12.150 | **0.007** |
| No | 455 (29.88) | 302 (27.50) | 80 (34.04) | 56 (40.00) | 17 (34.00) | | |
| Yes | 1068 (70.12) | 796 (72.50) | 155 (65.96) | 84 (60.00) | 33 (66.00) | | |
| PCI/CABG, n (%) | | | | | | $\chi^2$ = 185.692 | **<0.001** |
| No | 475 (31.19) | 233 (21.22) | 128 (54.47) | 80 (57.14) | 34 (68.00) | | |
| Yes | 1048 (68.81) | 865 (78.78) | 107 (45.53) | 60 (42.86) | 16 (32.00) | | |
| Thrombolysis use, n (%) | | | | | | - | 0.246 |
| No | 1518 (99.67) | 1095 (99.73) | 234 (99.57) | 140 (100.00) | 49 (98.00) | | |
| Yes | 5 (0.33) | 3 (0.27) | 1 (0.43) | 0 (0.00) | 1 (2.00) | | |
| Insulin use, n (%) | | | | | | $\chi^2$ = 3.837 | 0.280 |
| No | 455 (29.88) | 329 (29.96) | 75 (31.91) | 42 (30.00) | 9 (18.00) | | |
| Yes | 1068 (70.12) | 769 (70.04) | 160 (68.09) | 98 (70.00) | 41 (82.00) | | |
| Antiplatelet agents use, n (%) | | | | | | - | **<0.001** |
| No | 1515 (99.47) | 1097 (99.91) | 231 (98.30) | 140 (100.00) | 47 (94.00) | | |
| Yes | 8 (0.53) | 1 (0.09) | 4 (1.70) | 0 (0.00) | 3 (6.00) | | |
| Anticoagulation agents use, n (%) | | | | | | $\chi^2$ = 133.398 | **<0.001** |
| No | 981 (64.41) | 802 (73.04) | 107 (45.53) | 58 (41.43) | 14 (28.00) | | |
| Yes | 542 (35.59) | 296 (26.96) | 128 (54.47) | 82 (58.57) | 36 (72.00) | | |
| Statins use, n (%) | | | | | | $\chi^2$ = 30.978 | **<0.001** |
| No | 335 (22) | 206 (18.76) | 81 (34.47) | 32 (22.86) | 16 (32.00) | | |
| Yes | 1188 (78) | 892 (81.24) | 154 (65.53) | 108 (77.14) | 34 (68.00) | | |
| 30-day mortality, n (%) | | | | | | $\chi^2$ = 135.705 | **<0.001** |
| No | 1296 (85.1) | 1001 (91.17) | 157 (66.81) | 111 (79.29) | 27 (54.00) | | |
| Yes | 227 (14.9) | 97 (8.83) | 78 (33.19) | 29 (20.71) | 23 (46.00) | | |
| Follow-up duration, days, Mean ± SD | 26.91 ± 7.97 | 28.30 ± 6.03 | 23.12 ± 10.73 | 25.37 ± 9.47 | 18.71 ± 12.93 | F = 53.143 | **<0.001** |

F: ANOVA, $\chi^2$: Chi-square test, -: Fisher exact.

AMI: acute myocardial infarction, DM: diabetes mellitus, SD: standard deviation, NSTEMI: Non-ST elevation myocardial infarction, STEMI: ST elevation myocardial infarction, CS: cardiogenic shock, HR: heart rate, SBP: systolic blood pressure, DBP: diastolic blood pressure, RR: respiratory rate, SOFA: the Sequential Organ Failure Assessment, SAPS: the Simplified Acute Physiology Score, GCS: the Glasgow Coma Scale, CCI: the Charlson Comorbidity Index, WBC: white blood cell, HB: hemoglobin, RDW: red cell distribution width, Cr: creatinine, INR: international normalized ratio, PT: prothrombin time, BUN: blood urea nitrogen, Na: sodium, K: potassium, Mg: magnesium, PCI: percutaneous coronary intervention, CABG: coronary artery bypass graft.

sepsis, CS, weight, HR, SBP, DBP, RR, temperature, $SpO_2$, SAPS-II, GCS, WBC, platelet, HB, Cr, pH, BUN, bicarbonate, Na, chloride, Mg, ventilation use, vasopressor use, PCI/CABG, antiplatelet agents use, anticoagulation agents use, and statins use were also significantly different (all $P<0.05$).

## Association of longitudinal trajectory of blood glucose with 30-day mortality

We first screened the covariates associated with 30-day mortality in AMI patients with DM (S3 Table). The results showed that age, AMI types, urine output, sepsis, CS, HR, DBP, RR, SpO$_2$, SOFA, SAPS-II, CCI, WBC, platelet, RDW, Cr, INR, PT, BUN, bicarbonate, chloride, Mg, vasopressor use, PCI/CABG, insulin use, anticoagulation agents use, and statins use were significantly associated with 30-day mortality (all $P<0.05$).

Then we investigated the association of different trajectory classes with 30-day mortality in AMI patients with DM. As it shown in the Table 2, after adjusting for all the covariates, the average blood glucose level of $\geq$200 mg/dL was linked to higher risk of 30-day mortality compared with that of <140 mg/dL (HR = 1.80, 95%CI: 1.23–2.63). Also, comparing to patients with longitudinal trajectory of blood glucose of class 1, those with class 2 (HR = 2.52, 95%CI: 1.79–3.53) or with class 4 (HR = 3.53, 95%CI: 2.07–6.03) seemed to have higher risk of 30-day mortality.

In addition, according to the KM curves, among these 4 classes of longitudinal trajectory of blood glucose, patients with trajectory of blood glucose of the class 4 had the lowest survival probability, followed by that of the class 2 (Fig 2).

## Relationship between longitudinal trajectory of blood glucose and 30-day mortality in subgroups of age, gender and AMI types

The association of short-time longitudinal trajectory of blood glucose with 30-day mortality was further explored in different subgroups (Fig 3). Concretely speaking, on the basis of the Table 3, after adjusting for covariates, the associations of longitudinal trajectory of blood glucose of class 2 and class 4 with high risk of 30-day mortality were also significant in patients who aged $\geq$60 years old, female, and male (all $P<0.05$). Specially, longitudinal trajectory of blood glucose of class 4 was associated with higher risk of 30-day mortality in patients with NSTEMI (HR = 3.63, 95%CI: 1.81–7.28), whereas that of class 2, 3, and 4 were all linked to higher risk of 30-day mortality in patients with STEMI (all $P<0.05$).

**Table 2. Association of different trajectory classes with 30-day mortality in AMI patients with DM.**

| Variables | Model 1 | | Model 2 | | Model 3 | |
|---|---|---|---|---|---|---|
| | HR (95% CI) | *P* | HR (95% CI) | *P* | HR (95% CI) | *P* |
| Blood glucose levels (mg/dL) | | | | | | |
| <140 | Ref | | Ref | | Ref | |
| 140–200 | 0.87 (0.59–1.29) | 0.494 | 1.04 (0.70–1.54) | 0.840 | 1.28 (0.85–1.95) | 0.241 |
| $\geq$200 | 2.24 (1.60–3.13) | <**0.001** | 1.45 (1.03–2.04) | **0.031** | 1.80 (1.23–2.63) | **0.002** |
| Classes | | | | | | |
| 1 | Ref | | Ref | | Ref | |
| 2 | 4.35 (3.23–5.87) | <**0.001** | 2.10 (1.52–2.89) | <**0.001** | 2.52 (1.79–3.53) | <**0.001** |
| 3 | 2.54 (1.67–3.84) | <**0.001** | 1.33 (0.86–2.04) | 0.195 | 1.38 (0.87–2.18) | 0.170 |
| 4 | 7.22 (4.58–11.39) | <**0.001** | 3.04 (1.86–4.94) | <**0.001** | 3.53 (2.07–6.03) | <**0.001** |

AMI: acute myocardial infarction, DM: diabetes mellitus, HR: hazard ratio, CI: confidence interval, Ref: reference.

Model 1: crude model

Model 2: adjusted for vasopressor use, PCI/CABG, antiplatelet use, anticoagulation agents use, and statins use

Model 3: adjusted for age, AMI types, urine output, sepsis, CS, HR, DBP, RR, SpO$_2$, SOFA, SAPS-II, CCI, WBC, platelet, HB, RDW, Cr, pH, INR, PT, BUN, bicarbonate, Na, chloride, Mg, vasopressor use, PCI/CABG, antiplatelet agents use, anticoagulation agents use, and statins use.

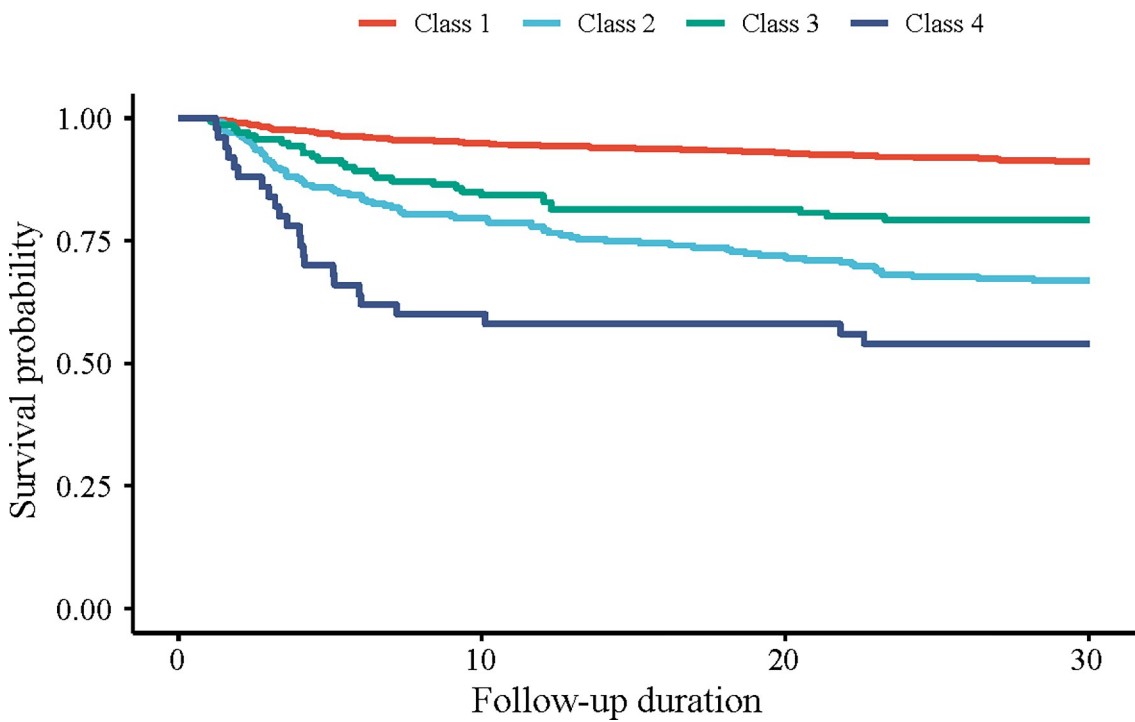

**Fig 2. The KM cumulative incidence curve of the occurrence of 30-day mortality in patients with different trajectory classes of blood glucose.**

## Discussion

The current research explored the association of short-term longitudinal trajectory of blood glucose with 30-day mortality in AMI patients with DM. The results showed that comparing to the average blood glucose level at ICU admission of <140 mg/dL, that of ≥200 mg/dL was linked to higher risk of 30-day mortality. Comparing to patients who had the longitudinal trajectory of blood glucose of class 1, those who had the longitudinal trajectory of blood glucose of class 2 or class 4 seemed to have higher risk of 30-day mortality. In addition, these associations between longitudinal trajectory of blood glucose of class 2 or class 4 and 30-day mortality were also found in subgroups of ≥60 years old, female, male, NSTEMI, and STEMI.

To the best of our knowledge, this study was the first to investigate the relationship between longitudinal trajectories of blood glucose and 30-day mortality in AMI patients with DM. Thoegersen et al. [17] suggested that patients with AMI complicated by CS and concomitant DM had a significantly higher 30-day mortality in comparison to patients without DM. High glucose levels on the ICU admission were associated with increased 30-day mortality in DM patients, which in a dose-dependent manner (4–8 mmol/L, 41%; 8–12 mmol/L, 49%; 12–16 mmol/L, 63%; >16 mmol/L 67%). Previous retrospective cohort studies showed that admission hyperglycemia was identified as an independent predictor of worse short-term and long-term outcomes in AMI patients with or without DM [5, 18]. Similarly, in present study, we observed that higher levels of average blood glucose at the ICU admission were significantly associated with increased risk of 30-day mortality in patients with AMI combined with DM (≥200 mg/dL, that is 11.2 mmol/L). Besides, Ishihara et al. [19] considered that there was a U-shaped relationship between glucose and mortality in patients with a history of DM, namely AMI patients with admission glucose 9–10 mmol/L had the lowest mortality, whereas lower glucose was better in nondiabetic patients. Compared with Ishihara's results, we found AMI

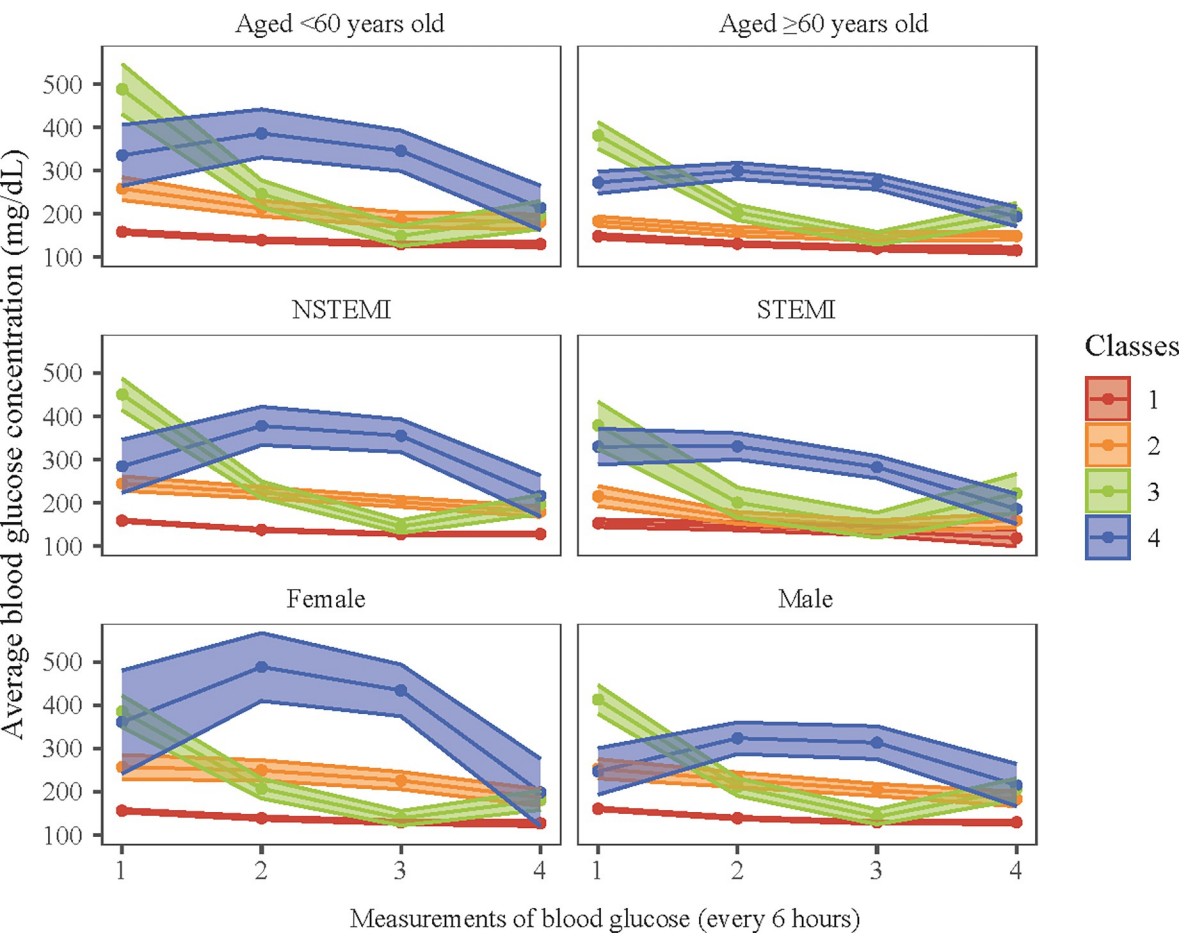

**Fig 3. The trajectory plot of 4 trajectory patterns of average blood glucose in subgroups of age, gender, and different AMI types.**

patients with DM who had the average admission blood glucose ≥200 mg/dL seemed to have higher risk of 30-day mortality instead of a U-shaped relationship between blood glucose and mortality risk. A possible reason may be the racial difference between the study populations, that Ishihara studied the Japanese persons whereas our study on the basis of population in the United States. Therefore, the causal relationship between admission blood glucose and short-term mortality in patients with AMI combined with DM needs to be further clarified.

The role of blood glucose trajectory in different diseases has been studied in recent decades, because glucose trajectories and trends can provide valuable real-time information on glucose conditions that may be a better target for clinical control than baseline blood glucose levels [20–22]. Li et al. [23] found that fasting plasma glucose trajectories were significantly associated with the lifetime risk of cardiovascular disease (CVD), and both decrease in fasting plasma glucose over time and consistently lower fasting plasma glucose over 4 years were linked to lower lifetime risk of CVD. Similarly, in the present study, we utilized LGMM model to determine different classes of short-term blood glucose trajectories, and found that comparing to patients who have a low average blood glucose level at the ICU admission with steady decreased trend, risk of 30-day mortality was not increased in those had a high average blood glucose level at the ICU admission with rapidly decreased trend in 24 hours after admission. Meanwhile, patients with not only a high level of average admission blood glucose but also a slowly reducing trend also had higher risk of short-term mortality. These results indicated that

**Table 3. Association of different trajectory classes with 30-day mortality in subgroups of age, AMI types, and gender.**

| Subgroup | Classes | HR (95% CI) | P |
|---|---|---|---|
| Aged <60 years old | 1 | Ref | |
| | 2 | 1.91 (0.77–4.73) | 0.164 |
| | 3 | 0.79 (0.27–2.37) | 0.680 |
| | 4 | 2.10 (0.52–8.56) | 0.299 |
| Aged ≥60 years old | 1 | Ref | |
| | 2 | 2.83 (1.91–4.17) | <**0.001** |
| | 3 | 1.48 (0.87–2.52) | 0.152 |
| | 4 | 3.14 (1.62–6.07) | <**0.001** |
| Female | 1 | Ref | |
| | 2 | 3.32 (1.79–6.16) | <**0.001** |
| | 3 | 1.50 (0.69–3.23) | 0.305 |
| | 4 | 2.95 (1.05–8.24) | **0.039** |
| Male | 1 | Ref | |
| | 2 | 2.16 (1.39–3.35) | <**0.001** |
| | 3 | 1.31 (0.71–2.43) | 0.385 |
| | 4 | 3.63 (1.81–7.28) | <**0.001** |
| NSTEMI | 1 | Ref | |
| | 2 | 1.33 (0.85–2.06) | 0.208 |
| | 3 | 0.79 (0.40–1.56) | 0.498 |
| | 4 | 2.39 (1.20–4.77) | **0.014** |
| STEMI | 1 | Ref | |
| | 2 | 6.39 (2.56–15.94) | <**0.001** |
| | 3 | 3.43 (1.20–9.86) | **0.022** |
| | 4 | 4.28 (1.11–16.58) | **0.035** |

AMI: acute myocardial infarction, HR: hazard ratio, CI: confidence interval, Ref: reference, NSTEMI: Non-ST elevation myocardial infarction, STEMI: ST elevation myocardial infarction.

Age subgroups: adjusted for AMI types, urine output, sepsis, CS, HR, DBP, RR, SpO$_2$, SOFA, SAPS-II, CCI, WBC, platelet, HB, RDW, Cr, PH, INR, PT, BUN, bicarbonate, Na, chloride, Mg, vasopressor use, PCI/CABG, antiplatelet agents use, anticoagulation agents use, and statins

Gender subgroups: adjusted for age, AMI types, urine output, sepsis, CS, HR, DBP, RR, SpO$_2$, SOFA, SAPS-II, CCI, WBC, platelet, HB, RDW, Cr, PH, INR, PT, BUN, bicarbonate, Na, chloride, Mg, vasopressor use, PCI/CABG, antiplatelet agents use, anticoagulation agents use, and statins

AMI types subgroups: adjusted for age, urine output, sepsis, CS, HR, DBP, RR, SpO$_2$, SOFA, SAPS-II, CCI, WBC, platelet, HB, RDW, Cr, PH, INR, PT, BUN, bicarbonate, Na, chloride, Mg, vasopressor use, PCI/CABG, antiplatelet agents use, anticoagulation agents use, and statins.

a quickly glucose-lowering treatment at early stage of the ICU could help reduce the mortality risk in AMI patients combined with DM, which may supplement the literature blank that association of short-term longitudinal trajectory of blood glucose with mortality risk in AMI patients with DM. However, although benefits of managing hyperglycemia such as intensive insulin therapy have been reported, the risk of hypoglycemia-related events should not be neglected. Therefore, application of faster and stricter glucose control should be more cautious in clinical. Besides, the stress hyperglycemia ratio (SHR) has been introduced to gain new insights into the relationship between hyperglycemia and patient outcomes by correcting glucose levels for glycosylated hemoglobin (HbA1c) in recent years [24]. Stress hyperglycemia has been proven to be a strong predictor of a higher risk of mortality and morbidity risk in patients

with AMI [25, 26]. According to both blood glucose trajectory and SHR has shown promise in predicting adverse events in CVD patients, would a composite score combining both acute hyperglycemia and longitudinal glucose trajectories offer additional insights in patients with AMI combined with DM may be interesting and imperative for further research.

The underlying mechanisms that deleterious effects of cardiac glucotoxicity results from a supra-physiological glucose insult to cardiomyocytes, cardiac endothelial cells, and the clotting system [27]. Hyperglycemia can reduce endothelium dependent vasodilatation and impairs endothelial repair, which has been thought to directly impact the remodeling of infarcted cardiac tissue, and directly magnify oxidative stress and inflammatory immune reaction in cardiac tissue following ischemia [28]. Besides, a diverse of perturbed metabolic pathways are involved in the AMI complicated with DM, including carbohydrate metabolism, glycolysis, lipid metabolism, amino acid metabolism and tricarboxylic acid cycle [29]. Reductions in some metabolites were negatively correlated with blood glucose and inflammatory markers, which might further compromise glucose expenditure and aggravate inflammation leading to poorer prognosis in DM-AMI [29]. Basing on the study results, we speculated that appropriate treatment taken to reduce and keep blood glucose at normal range at early stage of the ICU stay quickly may through improving hyperglycemia that associated with inflammation and metabolic abnormalities, thereby relieving the pressure of myocardial tissue remodeling, and further improve prognosis in patients with AMI combined with DM. Nevertheless, the specific mechanisms that different trajectories of blood glucose affecting short-term mortality in AMI combined with DM need to be further clarified.

When compared different indexes among patients with blood glucose trajectories of class 1 to 4, we found that those who in class 2 group and class 4 group had higher average levels of DBP and serum Na concentration. Also, more than half of them had not received PCI/CABG, and used anticoagulation agents. In fact, DBP has been demonstrated to be a useful parameter for prediction of long-term mortality in several populations including CVD [30, 31]. Terlecki et al. [32] considered that low difference between serum Na and chloride ion concentrations on admission was linked to an increased risk of 30-day mortality in patients with AMI treated with PCI and may serve as a useful prognostic marker. In the current study, we have adjusted these potential confounding factors in multivariate models, including DBP, Na, PCI/CABG, and so on. Therefore, the results were relatively robust.

In addition, the relationship between short-term trajectory of blood glucose and 30-day mortality was assessed in subgroups of age, gender and different AMI types. Patients with blood glucose trajectory of class 2 or class 4 having a higher risk of 30-day mortality was observed in aged ≥60 years old, female, male, and STEMI subgroups. Age is inversely proportional to survival in patients with AMI [33]. The ability to regulate blood glucose and the rate of cardiac muscle reconstruction may be decreased in patients aged ≥60 years old, and therefore, our findings indicated that age on its own should not be a reason to withhold rapid hypoglycemic measures use. It has shown that acute blood glucose elevation in patients with STEMI suggests a poor prognosis [26, 34, 35]. Interestingly, in patients with NSTEMI, only blood glucose trajectory of class 4 was linked to the risk of 30-day mortality. The class 4 of blood glucose trajectory was having a relative high level of admission blood glucose, and rising slowly and then had a falling trend. However, the specific therapeutic schedule of each participant could not be obtained from the database, which limited the inference about the possible causes of this difference. In summary, results of subgroup analysis supplemented that regardless of how high the baseline blood glucose level is, maintaining a downward trend in blood glucose for 24 hours after admission is critical in NSTEMI patients.

This study based on the MIMIC database, a large public clinical database, contains a large sample of real cases that is a representative population in the United States. Longitudinal

trajectory of blood glucose is a cheap and convenient index in clinical, and our findings may provide some references for further exploration on assistance of mortality risk monitoring and early warning in AMI patients with DM. However, there are still some limitations in the current research. Due to the retrospective cohort design, this study inevitably has certain selection bias. Potential confounding variables, such as lifestyle factors, myocardial infarction size, and other pathological information, are not available in the database, which limited the adjustment of covariates associated with prognosis and may confound the association of longitudinal trajectory of blood glucose with mortality risk in AMI patients with DM. Moreover, patients in the MIMIC database were from a single-center, and therefore, multicenter prospective cohorts are needed to clarify the causal association between longitudinal trajectories of blood glucose and DM-AMI prognosis.

## Conclusion

A low average blood glucose level at the ICU admission as well as taking appropriate measures to keeping blood glucose within a normal range at the early stage of the ICU stay may be beneficial to improve short-term prognosis in AMI-DM patients, but the precise range of blood glucose control need further clarification.

## Supporting information

**S1 Table. Determination of the number of classes in LGMM model.**
(DOCX)

**S2 Table. Average posterior probability of the selected classes in LGMM model.**
(DOCX)

**S3 Table. Covariates associated with 30-day mortality in AMI patients with DM.**
(DOCX)

**S1 Data.**
(XLSX)

## Author Contributions

**Conceptualization:** Bowen Deng.

**Data curation:** Qingcheng Liu, Liang Qiao, Shun Lv.

**Formal analysis:** Qingcheng Liu, Liang Qiao, Shun Lv.

**Investigation:** Qingcheng Liu, Liang Qiao, Shun Lv.

**Methodology:** Qingcheng Liu, Liang Qiao, Shun Lv.

**Writing – review & editing:** Bowen Deng.

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
