## [Decision Letter · Decision Letter 0]

8 Jul 2024

PONE-D-24-11576Longitudinal trajectories of blood glucose and 30-day mortality in patients with diabetes mellitus combined with acute myocardial infarction: a retrospective cohort analysis of the MIMIC databasePLOS ONE

Dear Dr. Deng,

Thank you for submitting your manuscript to PLOS ONE. After careful consideration, we feel that it has merit but does not fully meet PLOS ONE’s publication criteria as it currently stands. Therefore, we invite you to submit a revised version of the manuscript that addresses the points raised during the review process. Please submit your revised manuscript by Aug 22 2024 11:59PM. If you will need more time than this to complete your revisions, please reply to this message or contact the journal office at plosone@plos.org. Please include the following items when submitting your revised manuscript: 

We look forward to receiving your revised manuscript.

Kind regards,

Chiara Lazzeri

Academic Editor

PLOS ONE

Journal Requirements:

Reviewers' comments:

Reviewer's Responses to Questions

**Comments to the Author**

1. Is the manuscript technically sound, and do the data support the conclusions?

Reviewer #1: Yes

Reviewer #2: Yes

2. Has the statistical analysis been performed appropriately and rigorously? 

Reviewer #1: Yes

Reviewer #2: Yes

3. Have the authors made all data underlying the findings in their manuscript fully available?

Reviewer #1: Yes

Reviewer #2: Yes

4. Is the manuscript presented in an intelligible fashion and written in standard English?

Reviewer #1: Yes

Reviewer #2: Yes

5. Review Comments to the Author

Reviewer #1: Thank you for the opportunity to review the manuscript investigating the correlation between blood glucose trajectories and 30-day mortality among patients with diabetes mellitus and acute myocardial infarction. Despite the lack of demonstrable benefits in treating acute hyperglycemia during acute MI in this patient population, the study holds clinical significance. However, the current literature review on the subject is limited. It is suggested that the authors provide a succinct elaboration on the stress hyperglycemia ratio, an emerging marker for stress hyperglycemia, which has shown promise in predicting adverse events in both diabetic and non-diabetic patients (source: https://pubmed.ncbi.nlm.nih.gov/38042441/). Incorporating previous evidence on this marker and including such an analysis, contingent upon pertinent data availability, could significantly enhance the study's value. Furthermore, would a composite score combining both acute hyperglycemia and longitudinal glucose trajectories offer additional insights? These considerations need further elaboration in the manuscript and underscore the imperative for further research in this domain.

Abstract: Clarity is needed regarding the stratification of patients into classes, particularly concerning Class 4, presumably comprising individuals with elevated glucose levels. Providing this specification would enhance comprehension.

Table 1: Certain variables, such as marital status, lack relevance and should be omitted for precision.

Reviewer #2: The blood glucose reduction in class 3 was associated with better outcomes, and the authors conjecture that this was owed to a better glucose-lowering treatment. But there is no data about the received treatment, so this trajectory could have other causes. Therefore, the conclusion should be more cautious in suggesting faster and stricter glucose control. While several studies have shown the benefits of managing hyperglycemia (>200 mg/dL), the risk of hypoglycemia-related events when using intensive insulin therapy should not be neglected.

Basal characteristics of the patients in each longitudinal trajectory of blood glucose class were very different. It is possible that there were relevant non-measured variables that could introduce confounding. Although the potential for confounding was addressed, it should be highlighted and discussed further.

It would be useful for a non-statistician to have a clear explanation in the manuscript about latent growth mixture modeling and how it derives trajectory classes from the data.

6. PLOS authors have the option to publish the peer review history of their article (what does this mean?). If published, this will include your full peer review and any attached files.

Reviewer #1: No

Reviewer #2: **Yes: **Diego Costa

---

## [Author Response · Author response to Decision Letter 0]

10 Jul 2024

Responses to Reviewers

Dear Editor and reviewers,

We highly appreciate and thank the Editors and the reviewers for your time and valuable comments on this manuscript. We have carefully studied the comments and suggestions and revised our paper accordingly. The following are our point-by-point responses to the comments. We hope that the revisions are acceptable and that our responses adequately address the comments. Thank you for your consideration.

Responses to Reviewer #1’s comments

Thank you for the opportunity to review the manuscript investigating the correlation between blood glucose trajectories and 30-day mortality among patients with diabetes mellitus and acute myocardial infarction. Despite the lack of demonstrable benefits in treating acute hyperglycemia during acute MI in this patient population, the study holds clinical significance. However, the current literature review on the subject is limited. It is suggested that the authors provide a succinct elaboration on the stress hyperglycemia ratio, an emerging marker for stress hyperglycemia, which has shown promise in predicting adverse events in both diabetic and non-diabetic patients (source: https://pubmed.ncbi.nlm.nih.gov/38042441/). Incorporating previous evidence on this marker and including such an analysis, contingent upon pertinent data availability, could significantly enhance the study's value. Furthermore, would a composite score combining both acute hyperglycemia and longitudinal glucose trajectories offer additional insights? These considerations need further elaboration in the manuscript and underscore the imperative for further research in this domain. 

Response: Thank you for your valuable suggestion. We have read this article, and added relevant discussion in the revised manuscript. 

Abstract: Clarity is needed regarding the stratification of patients into classes, particularly concerning Class 4, presumably comprising individuals with elevated glucose levels. Providing this specification would enhance comprehension.

Response: We have clarified the details on 4 different classes of blood glucose trajectory both in the abstract section and method section.

Table 1: Certain variables, such as marital status, lack relevance and should be omitted for precision.

Response: Thank you for your valuable comment. We have removed certain variables including marital status and insurance in the Table 1.

Responses to Reviewer #2’s comments

The blood glucose reduction in class 3 was associated with better outcomes, and the authors conjecture that this was owed to a better glucose-lowering treatment. But there is no data about the received treatment, so this trajectory could have other causes. Therefore, the conclusion should be more cautious in suggesting faster and stricter glucose control. While several studies have shown the benefits of managing hyperglycemia (>200 mg/dL), the risk of hypoglycemia-related events when using intensive insulin therapy should not be neglected.

Response: Thank you for your valuable comment. We have revised the description of results. We are very agree that the conclusion should be more cautious in clinical suggestions, and therefore relevant content in results section and discussion section has been revised. 

Basal characteristics of the patients in each longitudinal trajectory of blood glucose class were very different. It is possible that there were relevant non-measured variables that could introduce confounding. Although the potential for confounding was addressed, it should be highlighted and discussed further. 

Response: Thank you for your valuable comment. We have added the discussion on potential confounding factors in the limitation section.

It would be useful for a non-statistician to have a clear explanation in the manuscript about latent growth mixture modeling and how it derives trajectory classes from the data.

Response: Thank you for your valuable suggestion. We have added more details on the LGMM model in the methods section.

---

## [Editor Report · Decision Letter 1]

15 Jul 2024

Longitudinal trajectories of blood glucose and 30-day mortality in patients with diabetes mellitus combined with acute myocardial infarction: a retrospective cohort analysis of the MIMIC database

PONE-D-24-11576R1

Dear Dr. Deng,

We’re pleased to inform you that your manuscript has been judged scientifically suitable for publication and will be formally accepted for publication once it meets all outstanding technical requirements.

Kind regards,

Chiara Lazzeri

Academic Editor

PLOS ONE
---

## [Editor Report · Acceptance letter]

31 Jul 2024

PONE-D-24-11576R1 

PLOS ONE

Dear Dr. Deng, 

I'm pleased to inform you that your manuscript has been deemed suitable for publication in PLOS ONE. Congratulations! Your manuscript is now being handed over to our production team.

Kind regards, 

on behalf of

Dr. Chiara Lazzeri 

Academic Editor

PLOS ONE